

# Study of marsh wetland landscape pattern evolution on the Zoigê Plateau due to natural/human dual-effects

Liqin Dong[1], Wen Yang[1], Kun Zhang[2], Shuo Zhen[3], Xiping Cheng[1] and Lihua Wu[1]

[1] School of Geography and Ecotourism, Southwest Forestry University, Kunming, China
[2] National Plateau Wetlands Research Center, Southwest Forestry University, Kunming, China
[3] School of Geographical Sciences, Northeast Normal University, Changchun, China

## ABSTRACT

Zoigê Plateau, China's largest plateau marsh wetland, has experienced large-scale degradation of the marsh wetland and evolution of the wetland landscape pattern over the past 40 years due to climate warming and human activities. How exactly do the wetland landscape pattern characteristics change? How do climatic change and human activities affect the wetland evolution? These questions are yet to be systematically investigated. In order to investigate changes to the marsh wetland on the Zoigê Plateau, field investigations, spatial and statistical analysis were undertaken. Findings from our study indicate that from 1977–2016, the area of marsh wetland on the Plateau reduced by 56.54%, approximately 66,700 hm$^2$ of marsh wetland has been lost. The centroids of both marsh and marshy meadow migrated and the landscape centroid migration behaviors were also correlated with the distribution and variation of the marsh wetland on different slopes. In addition, the number of marsh landscape patches initially increased before decreasing; the number of marshy meadow landscape patches also recorded an initial increase, followed by a decline before a final increase. As the effects of human activities weakened, the aggregation degrees of both marsh and marshy meadow increased. Overall, the fragmentation degree, diversity and fractal dimension of the marsh wetland all declined. An investigation into the driving factors affecting the Plateau area shows that the increase of annual average temperature was the natural factor while trenching and overgrazing were the main human factors resulting in wetland degradation. Results from this study provide basic data and theoretical foundation for the protection and restoration of marsh wetland in alpine regions.

Corresponding authors
Liqin Dong, dongliqin@swfu.edu.cn
Kun Zhang, zhangkun@swfu.edu.cn

## INTRODUCTION

Since the 1950s, global temperatures have experienced unprecedented warming trends due to changes in the global climate system; recent climate change has imposed extensive impacts on human and natural systems (*IPCC, 2014*). Climate warming has accelerated permafrost melting (*Grosse et al., 2011*), and increased the degradation of wide-range
plateau marsh and marshy meadows (*Xiang, Guo & Wu, 2009*), resulting in the ecological environment in alpine regions to be vulnerable to increasing temperatures (*Zhao et al., 2018*).

Landscape pattern indices is an important means in landscape pattern analysis (*Chen et al., 2008*). For effective monitoring and protection, many studies analyzed the dynamic changes of wetlands over time, and the underlying factors responsible have also received attentions (*Liu, 2019*; *Cong et al., 2019*; *Yu et al., 2017*). The Qinghai-Tibet Plateau, known as the "Third Pole", is a sensitive indicator of global change, whose unique natural conditions (namely, high altitude and cold weather) determine the vulnerability of the plateau ecosystem. This ecosystem can generate intensive responses to even tiny environmental fluctuations (*Klein, Harte & Zhao, 2004*). The Zoigê Plateau, located on the eastern edge of the Qinghai-Tibet Plateau, is a marsh wetland having the highest altitude and greatest area globally. This area is also the most extensively-distributed marsh wetland in China. The Zoigê Plateau is also an important water conservation and biodiversity gathering region for the Yangtze and Yellow Rivers (*Hu et al., 2015*; *Liu et al., 2019*). Unique climatic, geological and hydrological conditions can provide a favorable living environment for wild animals and plants. In addition to being an important habitat of *Grus nigricollis* (black-necked crane; the first-class protected animal in China), the Qinghai-Tibet Plateau is also one of five pastures in China (*Wang et al., 2008*). The Zoigê Plateau is therefore very important for the sustainable development of the regional environment, ecology, society and economy, and it imposes significant effects on water resource safety in the whole Yangtze and Yellow River basins.

However, due to the combined effects of the natural environment and human activities, the marsh wetland landscape on the Zoigê Plateau has exhibited significant changes over time. The marsh and marshy meadow have experienced noticeable degradation; the area of marsh has recorded a sharp reduction and the marshy meadow has become fragmented (*Zhen et al., 2017*). Although changes are related to the combined effects of climatic and hydrological factors under natural-human dual-action, the detailed driving mechanism of the evolution of the wetland landscape pattern, especially analysis of typical marsh and marshy meadow, is still unclear.

This study focused on typical marsh and marshy meadow on the Zoigê Plateau, and systematically investigated the evolutionary tendency of the plateau marsh wetland landscape pattern to examine the effects of climatic change and human activities on wetland evolution. Gaining in-depth knowledge of the Zoigê marsh wetland can not only provide essential data for investigating the evolution of typical marsh wetland ecosystem on the Plateau, it can also provide an important theoretical foundation for the protection and sustainable development of the Zoigê Wetland by exploring the driving mechanism of wetland landscape pattern evolution.
## EXPERIMENTAL DESIGN AND RESEARCH METHOD

### The research area

Zoigê County is located in the northwestern area of Sichuan, China (102°08′E∼103°39′E and 32°56′N∼34°19′N). Zoigê Plateau is located in the northeastern area of the Qinghai-Tibet Plateau. Zoigê Wetland is not only the highest-altitude marsh wetland in the world, having an altitude of approximately 3,200∼3,700 m, it has the greatest area and most extensive distribution in China. This marsh area is therefore representative of the alpine wetland ecosystem on the Qinghai-Tibet Plateau (Fig. 1).

### Data source

Landsat MSS images from 1977, Landsat TM images from 1994 and 2007, and Landsat OLI images from 2016 were collated (Table 1), and a digital topographic map of the research area (with a resolution of 90 m) was used. Specifically, MSS images at the optimal 6th, 5th and 4th wavebands, TM images at 5th, 4th and 3rd wavebands and OLI images at 5th, 6th and 2nd wavebands were used.

### Data processing

After geometric correction, radiometric calibration and atmospheric correction, each satellite image was stitched and cropped. All images were finally projected using WGS_1984_UTM_ZONE_48N. We employed manual visual detection for the interpretation of remoting-sensing images. By referring to previous experiences and practical conditions, the interpretation signs of different types of wetlands were established (as listed in Table 2). According to the established interpretation signs, wetlands with an area greater than 0.81 hm$^2$ (excluding permanent and seasonal rivers and riverbeds) were preliminarily extracted by combining supervised classification and manual visual interpretation. In order to display the changes of two types of marsh wetlands, we uniformly classified grasslands, mountainous regions, cities and rivers & lakes as unclassified type. At each investigation stage, practical conditions in the local marshy wetland, as well as trenching and grazing results for the last three years, were analyzed using visual inspection. More than 100 field investigation points were also used which were uniformly distributed on the Zoigê Plateau. Based on interpretation results, the topographic map of the research area and field investigation points, landscape patches were manually modified and supplemented.

Additionally, a number of indices including landscape fragmentation, landscape diversity, landscape uniformity and landscape fractal dimension were used to analyze overall landscape change. These indices were defined as:

(1) Landscape fragmentation, reflecting the landscape's segmentation degree, can be characterized by the value of PD. Landscape fragmentation is equal to the number of patches divided by total area:

$$\mathrm{PD} = Ni/Ai \tag{1}$$

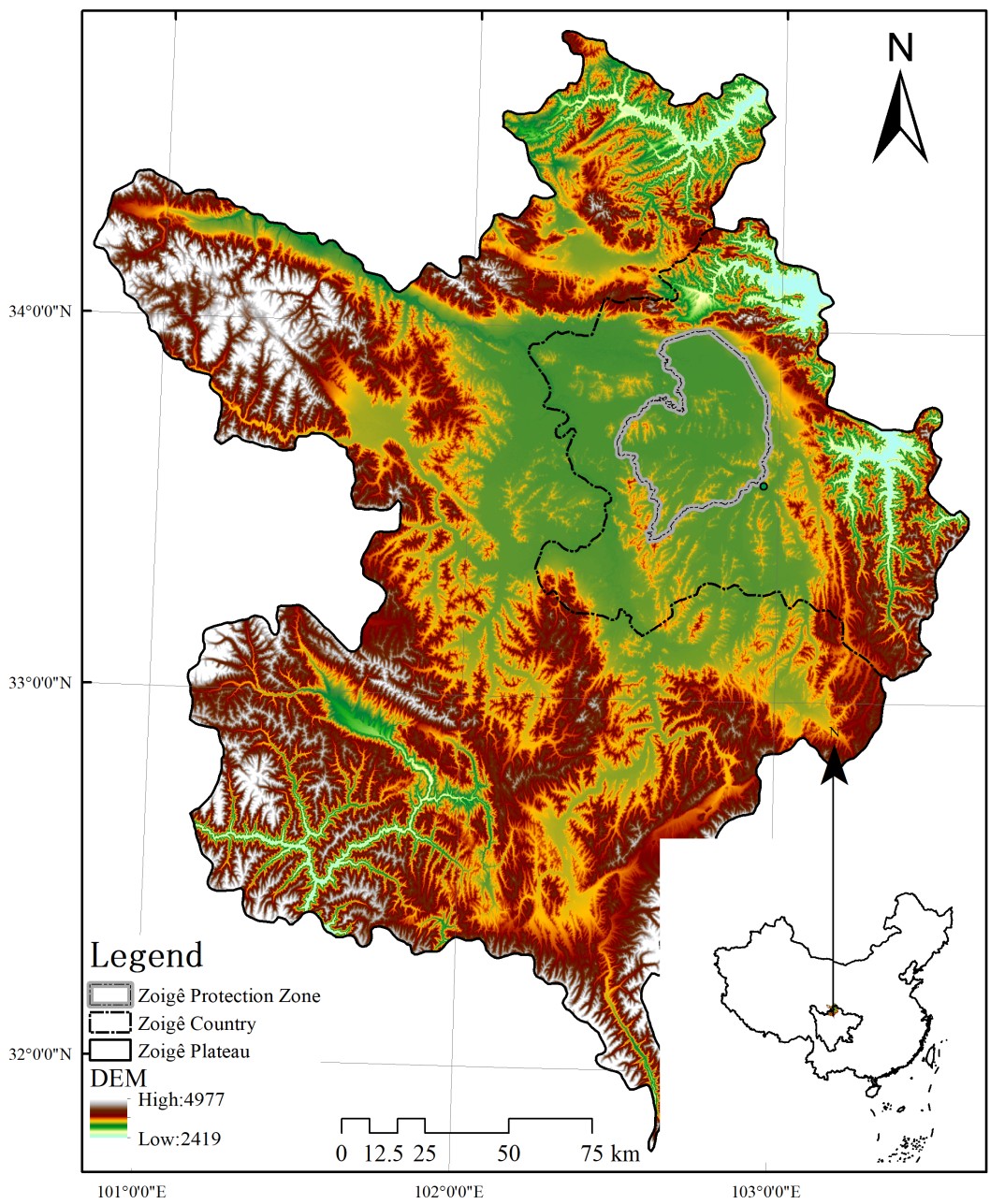

**Figure 1** **Location of the Zoigê Plateau wetland.**

where, PD denotes landscape fragmentation; $Ni$ denotes the number of landscape patches; and $Ai$ denotes the total area of this type of landscape. Landscape fragmentation

(2) Landscape diversity, denoted as $H$, can be calculated as:

$$H = -\sum_{k=1}^{m} P_k \ln P_k \qquad (2)$$

**Table 1  Landsat images under in this study.**

| Landsat MSS | | Landsat TM | | Landsat TM | | Landsat OLI | |
|---|---|---|---|---|---|---|---|
| Tract number | Data | Tract number | Data | Tract number | Data | Tract number | Data |
| 140036 | 1977.9.18 | 130037 | 1994.6.26 | 130037 | 2007.9.18 | 130037 | 2016.6.22 |
| 140037 | 1977.9.25 | 131036 | 1994.8.04 | 131036 | 2007.9.25 | 131036 | 2016.7.15 |
| 141036 | 1977.9.25 | 131037 | 1994. 7.03 | 131037 | 2007.9.25 | 131037 | 2016.7.15 |
| 141037 | 1977.9.25 | 131038 | 1994.9.05 | 131038 | 2007.9.25 | 131038 | 2016.7.15 |
| 142036 | 1977.9.24 | 132036 | 1994.8.27 | 132036 | 2007.9.24 | 132036 | 2016.6.20 |
| | | 132037 | 1994.10.14 | 132037 | 2007.9.24 | 132037 | 2016.6.20 |

**Table 2  Image example showing part of the object type.**

| Feature name | Marsh | Marshy meadow | Meadow | Mountain | Town |
|---|---|---|---|---|---|
| Image example | | | | | |

where, $P_k$ denotes the proportion of the area of the $k$th type of landscape in the total area of all types of landscapes; and $m$ denotes the number of landscape types in the research area. Equal proportions of the areas of different types of landscapes in the total research area suggest high landscape diversity. In contrast, if the proportions of the areas of different types of landscape in the total research area vary significantly, the landscape exhibits low diversity. The value of $H$ therefore reflects landscape diversity.

(3) Landscape uniformity, denoted as E, can be calculated as:

$$E = (H/H_{max}) \times 100\% \tag{3}$$

where, $H = -\sum_{k=1}^{m} P_k \ln(P_k)$; and $H_{max} = \ln(m)$. Landscape uniformity reflects the distribution uniformity of different types of landscapes and is reversely proportional to the index of the landscape dominance index. The sum of landscape uniformity and landscape dominance is equal to 1, i.e., these two indices can confirm with each other.

(4) Landscape fractal dimension, denoted as FRAC_AM, can be calculated as:

$$FRAC\_AM = 2\ln(P/4)\ln A \tag{4}$$

where, FRAC_AM denotes the fractal dimension, with a theoretical range of 1~2; $P$ denotes the perimeter of the patch; and A denotes the area of the patch. The fractal dimension of a patch is always used for measuring patch shape and the complexity of the patch edge, which can reflect the interference degree of human activities on the natural landscape to a certain degree. A larger value of landscape fractal dimension is indicative of a more regular landscape shape, whilst a landscape with a larger value of fractal dimension is more subjected to human activities.

## RESULTS

### Variation of landscape area of the marsh wetland

After interpretation of the remote-sensing images, interpretation precision was evaluated using the established confusion matrix. ROI data generated using over 100 field investigation points were used as the test data source. According to the test results, overall classification precision was 95.9617% and the calculated kappa coefficient was 0.9182, suggesting favorable classification results. Based on statistical analysis and comparison of the interpretation results, variations of marsh wetland from 1977 to 2016 were plotted (Figs. 2 and 3). According to the visual interpretation results of remote-sensing images, the marsh wetland area on the Zoigê Plateau reduced by 113,719 hm$^2$ from 1977 to 2016, with a mean rate of decrease of 2,842 hm$^2$/a. Despite a continuous reduction in area, the degradation rate of the marsh wetland significantly reduced, and the mean rate of decrease from 2007 to 2016 was only 1,658 hm$^2$/a. Currently, the marshy meadow significantly exceeds the marsh in terms of landscape area, which is still the main type of marsh wetland. Degraded marsh has mainly evolved into marshy meadow while degraded marshy meadow has mainly become a meadow landscape, confirming the general degradation direction: marsh - marshy meadow - meadow.

### Centroid migration of the marsh wetland on the Zoigê Plateau

Using ArcGIS software, the centroid coordinate was calculated based on the interpretation results. Remote sensing images were projected into the WGS 1984 UTM Zone 48N coordinate system and the centroids of two types of marsh wetlands were calculated. The related centroid migrations were extracted and plotted utilizing the statistical functions of the $X$- and $Y$-coordinates of the centroid in the attribute list (Figs. 4 and 5). The migration orientations, angles and distances of these two types of marsh wetlands were calculated using ArcGIS 10.2 functional module (COGO).

Overall, the centroid of the marsh landscape underwent reverse migration, initially recording southwestward migration before a northeastward migration. From 1977 to 1994, the centroid of the marsh landscape exhibited a migration of 6.16 km towards the west by south (WbS) by 27.40°, after which the centroid of the marsh landscape moved by 11.79 km towards the south by west (SbW) by 30.22°. Due to increasing demands on pasture, people were engaged in large-scale trenching and drainage activities on the Zoigê Plateau. In addition, deforestation in the western mountain forest led to the decline in the conservation function of the wetland water source, thereby resulting in a sharp reduction of the wetland area in the northeastern area. Interpretation results indicate that marsh landscape in the northeast part degraded significantly and the centroid constantly migrated towards the southwest.

In combination with the interpretation results and the general evolution direction of the marsh wetland degradation, the degraded marsh landscape predominantly evolved into a marshy meadow landscape, therefore the centroid of the marshy meadow landscape exhibited a migration direction that was almost completely opposite to that of the marsh landscape. The area of marshy meadow gradually moved towards the northeast direction during the period from 1977 to 2007. From 1977 to 1994, the centroid of the marshy

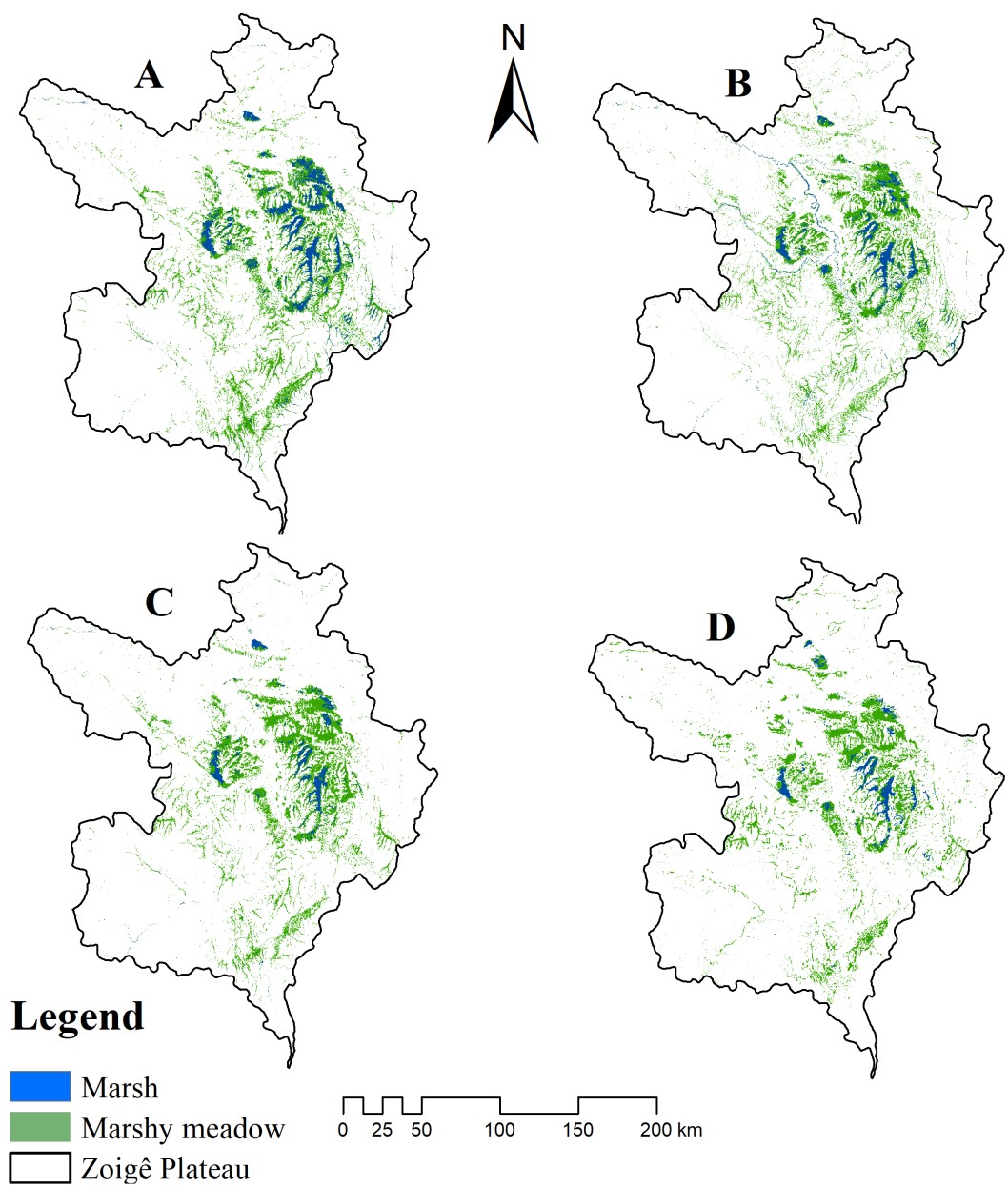

**Legend**

■ Marsh
■ Marshy meadow
□ Zoigê Plateau

0  25  50    100    150    200 km

**Figure 2  Distribution change of marsh wetland in the Zoigê Plateau over time.** (A) 1977, (B) 1994, (C) 2007 and (D) 2016.

meadow landscape moved by 10.73 km towards the north by west (NbW) by 21.03°. During the period from 1994 to 2007, the centroid continued moving towards the NbW by 2.88°, with a migration distance of 20.26 km, and the migration was almost in a true north direction. The marshy meadow landscape in the Zoigê Protection Zone on the central Plateau can be well protected, and the degraded wetland was partly restored, thereby leading to significant migration of the marshy meadow (15.13 km) towards the SbW by 24.44°.
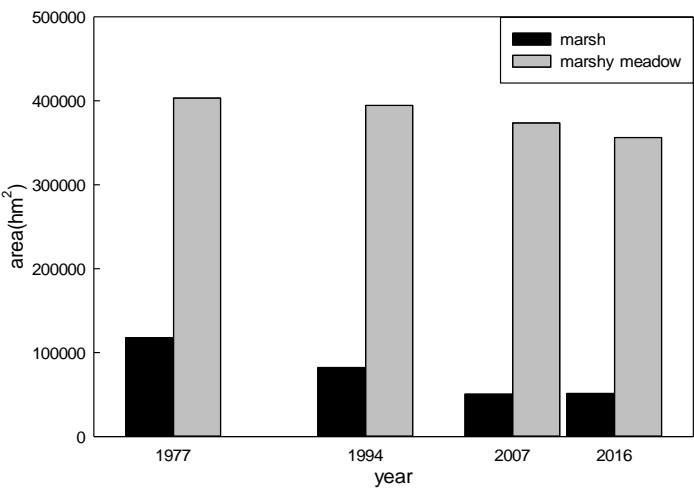

**Figure 3   Changes in marsh wetland landscape areas in the Zoigê Plateau over time.**

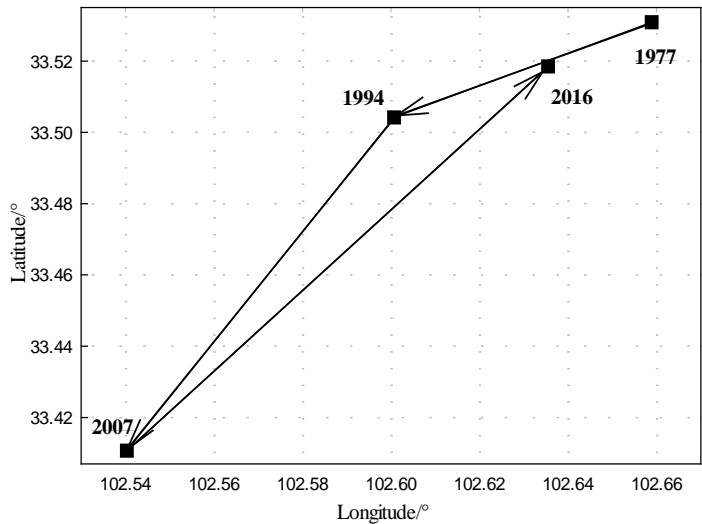

**Figure 4   The migration situation of the centroid of the marsh.**

## Variations of marsh wetlands on different slopes

Based on original digital DEM data of the research area with a resolution of 90 m, slope data were generated using the ArcGIS10.2 platform. Slope data was then spatially superimposed with the interpreted classification results of the marsh wetland in the research area to examine area distribution of the marsh wetland under different slope ranges.

Due to water catchment and holding properties, marsh wetland is always distributed in an area with a small slope. In this study, areas with a slope greater than 10° were uniformly classified as a type for calculation. According to statistical results in 2016, 90% of marsh wetland was distributed in the region with a slope angle less than 4° and 95% of marsh

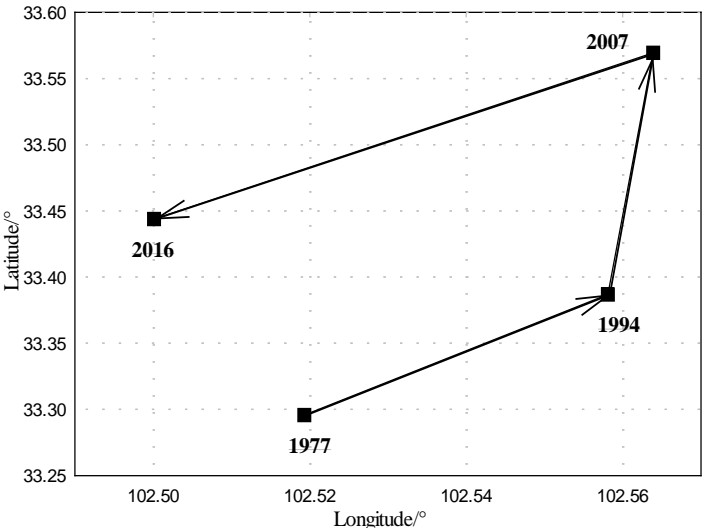

**Figure 5 The migration situation of the centroid of the marshy meadow.**

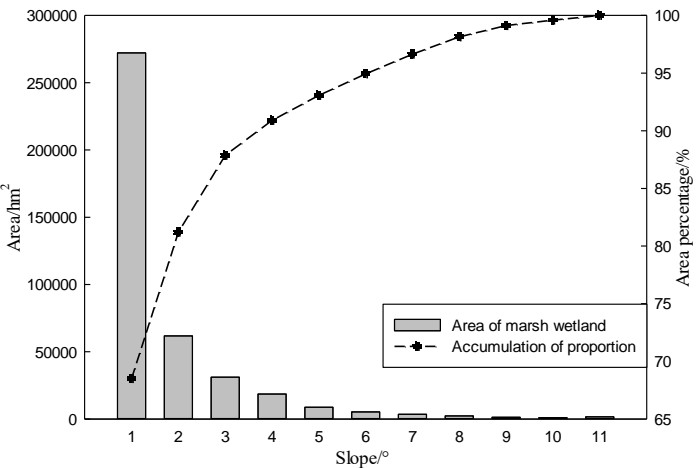

**Figure 6 Distribution of marsh wetland with different slope gradients in 2016.**

wetland was distributed in the region with a slope less than 7°. The distribution of marsh wetland in areas with different angles of slope for 2016 and all time periods are shown in Figs. 6 and 7 respectively.

The reduction of marsh wetland area in regions with different angles of slope from 1977 to 2016 is shown in Figs. 8. It can be seen that as the topographic slope of the research area increased, the area of marsh wetland significantly increased, i.e., the marsh wetland area exhibited a positive correlation with topographic slope ($r = 0.97$, $n = 10$ and $p < 0.01$). In other words, in the region with a larger topographic slope, both degradation area and
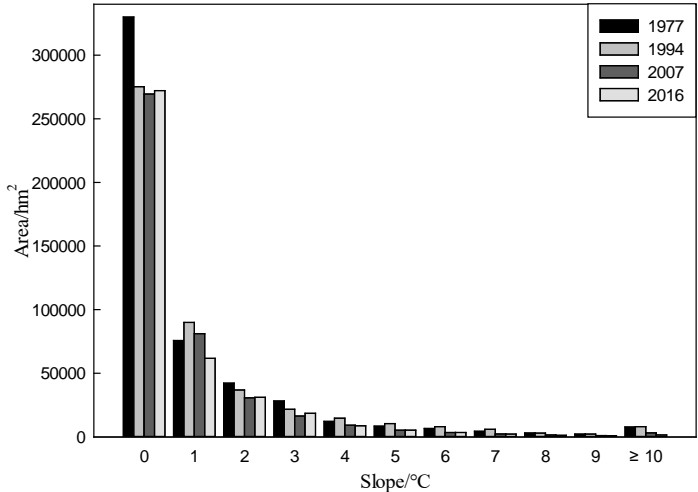

**Figure 7  Area chart of wetland in the study area with different slope gradients in the four time periods.**

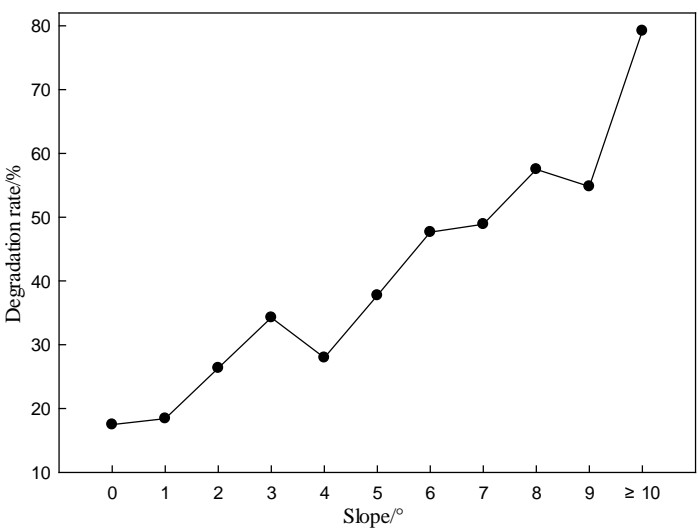

**Figure 8  The reduction rate of wetland in different slope gradients, 1977–2016.**

intensity of the marsh wetland were greater. This finding also reflects the spatial migration characteristics of the marsh wetland. The marsh wetland landscape constantly moved downward towards local valleys and gentle slopes, and tended to be concentrated on the bottom of the gentle hills. At the same time, the area of the marsh wetland on the slope also decreased.

**Table 3   The number of landscape patches in the Zoigê Plateau for the four time periods.**

| Year | Marsh wetland type | Number of landscape patches | Average patch area (hm²) |
|---|---|---|---|
| 1977 | Marshy meadow | 7666 | 13.14 |
| | Marsh | 2427 | 12.14 |
| 1994 | Marshy meadow | 9984 | 12.15 |
| | Marsh | 4671 | 4.39 |
| 2007 | Marshy meadow | 6486 | 14.37 |
| | Marsh | 1381 | 9.12 |
| 2016 | Marshy meadow | 9577 | 9.34 |
| | Marsh | 1014 | 12.66 |

## Variation of the marsh wetland landscape pattern indices on the Zoigê Plateau

### Patch scale

The patch numbers of marshy meadow and marsh landscapes on the Zoigê Plateau from 1977 to 2016 are shown in Table 3. Overall, the number of patches in the marshy meadow recorded an initial increase, a decrease and then a final increase. The internal marsh wetland change can be described below. The number of patches in the marshy meadow significantly increased. The newly-added patches were mainly small patches with an area less than 10 hm². Accompanied with an overall decreasing marsh wetland area, the marshy meadow became more fragmented. During the period from 1977 to 1994, the number of patches in the marsh landscape increased and the mean patch size decreased. After 1994, the number of patches declined and the mean patch size increased, while the marsh landscape tended to be distributed in continuous and a centralized pattern until 2016.

### Landscape type scale

In terms of landscape type scale, two types of landscapes (marsh and marshy meadow) exhibited different landscape pattern characteristics. The marsh landscape tended to be concentrated while the marshy meadow was gradually fragmentized, which can also be confirmed by the above patch scale results. The correlation between different landscape pattern scales can also be reflected.

(1) Marsh landscape

Table 4 lists the variation tendency and ecological significance of each marsh landscape pattern index. According to the variation tendencies of total area (TA) and mean patch size (MPS), the marsh landscape patches underwent perforation, segmentation and shrinkage fragmentation from 1977 to 1994. From 1994 to 2007, overall marsh landscape degraded and multiple patches were connected and merged. Finally, TA increased and the patches were further connected from 2007 to 2016.

From the variation tendencies of the proportion of patch in overall landscape area (PLAND) and the proportion of the greatest patch in the landscape area (LPI), it can be observed that the proportion of marsh wetland patches in total landscape area steadily declined from 1977 to 2007, but increased from 2007 to 2016. These results are in accordance

**Table 4  Landscape pattern index of the marsh wetland.**

| Index | TA | MPS | PLAND | LPI | LSI | IJI | COHESION | AI |
|-------|--------|-------|-------|------|-------|-------|----------|-------|
| 1977 | 117967 | 12.14 | 2.77 | 0.29 | 74.45 | 40.11 | 98.18 | 87.13 |
| 1994 | 82209 | 6.39 | 1.93 | 0.29 | 87.14 | 87.99 | 97.51 | 81.90 |
| 2007 | 50520 | 9.12 | 1.18 | 0.26 | 44.20 | 56.24 | 97.75 | 88.41 |
| 2016 | 51258 | 12.66 | 1.20 | 0.26 | 29.92 | 60.57 | 97.97 | 92.31 |

with the change in marsh landscape area. The stepwise drop of LPI reflected that significant degradation of the Zoigê Marsh Wetland (with the greatest marsh area on the Plateau) and almost unchanged landscape area in the previous and next stages.

A larger landscape shape index (LSI) is indicative of a more regular shape of this type of patch. The practical significance of LSI is always subjected to human activities. In the research area, LSI initially increased and then decreased, suggesting that the effect of human activities on marsh landscape type was initially enhanced and then weakened. The observation was also consistent with human activities on the Zoigê Plateau, i.e., protection followed by development.

Interspersion juxtaposition index (LJI) reflects the adjacent degree of this type of patch with the other types of patches. A larger LJI suggests a higher adjacent degree. The LJI value also represents the degree of influence terrain and hydrological effects have on this type of patch. The calculated LJI of the marsh landscape on the Zoigê Plateau exhibited three various phases. The differences among the different phases were significant, suggesting that the marsh landscape was heavily subjected to moisture and was almost unaffected by terrain. The marsh with better hydrological conditions was more likely to border upon the other patches.

The landscape cohesion index (hereinafter referred to as COHESION) reflects the connection degree between two patches. As shown in Table 4, the connection degree of the marsh landscape was almost maintained at a fixed level, which only slightly declined before increasing. The landscape aggregation index (AI) increased, i.e., the patches tended to be connected and exhibited continuous distribution and the landscape was only segmented and separated from 1977 to 1994. The variation tendencies of COHESION and AI were in complete conformity with the variation tendencies of TA and MPS of the marsh.

(2) Marshy meadow landscape

Results for marshy meadow landscape pattern indices (Table 5) indicate that TA and PLAND declined gradually, LPI and LSI initially increased before declining, , IJI declined steadily, COHESION was almost unchanged, AI and MPS initially declined before increasing.

Through comprehensive analysis of the eight landscape pattern indices, marshy meadow exhibited similar variation tendency with the marsh. The marshy meadow and marsh differed in that the marshy meadow landscape underwent more complex changes over time, and some indices exhibited three-phase variation tendencies.

The change of TA and PLAND revealed that the area of marshy meadow steadily declined. According to the variation tendency of MPS, the marshy meadow experienced

**Table 5   Landscape pattern index of the swamp meadow.**

| Index | TA | MPS | PLAND | LPI | LSI | IJI | COHESION | AI |
|-------|--------|-------|-------|------|--------|-------|----------|-------|
| 1977 | 403302 | 13.14 | 9.48 | 0.57 | 212.48 | 66.77 | 98.89 | 79.98 |
| 1994 | 394498 | 8.15 | 9.38 | 1.03 | 271.83 | 47.46 | 97.97 | 74.23 |
| 2007 | 373618 | 14.36 | 8.77 | 1.22 | 153.27 | 44.34 | 98.98 | 85.01 |
| 2016 | 356292 | 9.34 | 8.41 | 0.63 | 112.03 | 28.92 | 98.18 | 88.84 |

fragmentation, connection and fragmentation in three different periods. The change of LPI demonstrated that the degradation velocity of the marsh patch into the largest marshy meadow between 1977 and 1994 exceeded the degradation velocity of the marshy meadow into meadow, thereby resulting in the increase of the largest marshy meadow area. However, LPI recorded a rapid decline from 2007 to 2016, suggesting that the patch was involved in fragmentation or artificially divided. The steady reduction of IJI reflected that the marshy meadow patch was closer to the other patches, which then resulted in the decline in landscape heterogeneity, i.e., the landscape was uniform. The value of COHESION still exhibited slight change, with a difference of less than 1%. As stated above, AI first decline before increasing, suggesting that the marshy meadow patch was similar to the marsh patch. Both marshy meadow and marsh patches tended to be separated and fragmented from 1977 to 1994, after which they were connected and centralized.

### Overall landscape pattern scale

In order to investigate the overall plateau landscape pattern (Fig. 9), the three main landscape pattern indices of the Zoigê Plateau in different phases were analyzed. Results show that both landscape pattern diversity (SHDI) and landscape pattern uniformity (SHEI) of the marsh wetland landscape on the Zoigê Plateau declined, while the landscape fragmentation and landscape fractal dimension (FRAC_AM) initially increased before decreasing. This finding suggests that the patch distribution tended to be more centralized and irregular. A certain difference between landscape type scale and patch scale was also evident. The marshy meadow became more fragmented while the fragmentation degree of the marsh declined, however the overall landscape tended to be more centralized. These results on the three scales are reasonable and correlative, therefore indicating a good interpretation of the landscape's ecological significance.

### Analysis of the driving force of the change of marsh wetland on the Zoigê Plateau

Landscape pattern changes on the Zoigê Plateau are subjected to the dual influences of climate change and human activities. Among all natural factors, climatic factors include factors affecting local landscape pattern. Therefore, three representative climatic factors were selected and their correlations with landscape pattern change indices were analyzed. In terms of human activities, local investigations were combined with previous studies for summarization.
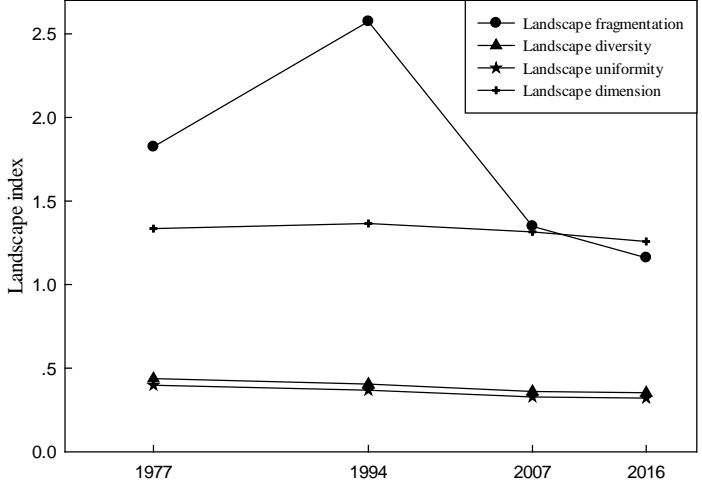

**Figure 9** **Change trend chart of landscape index of the Zoigê Plateau.**

**Table 6** **Correlation coefficients of climate factors and landscape pattern index.**

|  | Area | Landscape fragmentation index | Landscape diversity index | Landscape uniformity index |
|---|---|---|---|---|
| Annual average temperature | 0.022[*] | 0.032[*] | 0.013[*] | 0.027[*] |
| Annual precipitation | 0.385 | 0.283 | 0.418 | 0.397 |
| Potential evapotranspiration | 0.232 | 0.242 | 0.267 | 0.220 |

**Notes.**

*level of significance 0.05.

## *Climatic factors*

Antecedent research results show that the temperature on the Zoigê Plateau was on the rise, rainfall gradually declined, and evaporation increased during the past decades, i.e., the climate on the Zoigê Plateau was gradually becoming warmer and dryer. In particular, the climate exhibited a sudden change in 1997 and became warmer significantly afterwards, while rainfall changed suddenly in 1985, declined after 1991 and continued to decrease until 2014 (*Zhen et al., 2016*).

For statistical analysis, multiple data of the research area in four different phases was selected to examine correlations between climatic factors and landscape pattern indices. Table 6 lists 12 sets of bi-variable correlation analysis results between three climatic factors and four landscape pattern indices.

As shown in Table 6, annual average temperature exhibits significantly negative correlations with marsh wetland area, landscape fragmentation degree, landscape diversity and landscape uniformity, and the correlation coefficients all passed the significance tests at a significance level of 0.05. Accordingly, it can be concluded that the change of annual average temperature greatly affected the reduction of marsh wetland area, the fragmentation of the landscape and the decline in landscape diversity. From an ecological
perspective, increasing temperature directly led to the change of hydrological conditions of the marsh wetland. Therefore, the variation tendency of temperature fit well with the change of marsh wetland area in the research area, i.e., the marsh wetland landscape pattern became more unitary and concentrated. In contrast, annual average rainfall data and annual latent evapotranspiration data were found to have less correlation with various landscape pattern indices (at a significance level of $p > 0.05$). However, this does not mean that the change of marsh wetland landscape was irrelevant to rainfall and latent evaporation. Hydrologic condition is a key factor that directly affects the evolution of marsh wetland. Both rainfall and latent evapotranspiration exhibited unstable annual changes, which sometimes increased and sometimes decreased, changes which had a significant effect on correlation analysis results.

The change of climatic factors can account for the reduction of marsh wetland area and the fragmentation of landscape pattern from 1977 to 2007, however they do not account for the increase of marsh landscape area from 2007 to 2016. Accordingly, in addition to climatic change, human activities also affected marsh wetland area and landscape pattern change.

### Human activities

In most cases, human activities impose effects on the hydrological cycle by changing coverage type and utilization structure on the land surface (*Schulze & Roland, 2000*). In addition, human activities can also affect the exploitation (*Scanlon et al., 2007*) and utilization of wetland water resources (*Kingsford, 2000*), including agricultural irrigation water and the construction of hydraulic projects, thereby imposing effects on the hydrological process and landscape pattern of the wetland. Among all human factors, excavation for drainage and overgrazing are two important driving factors on the Zoigê Plateau.

Since 1970, the area of pasture has increased due to the excavation of drainage ditches on the Zoigê Plateau. With an increase in the local population and the development of animal husbandry (it is well known that animal husbandry serves as the major economic industry on the Zoigê Plateau, which takes up over 90% of gross agricultural production in Zoigê County), herdsmen began to strengthen grazing and the demand on meadow has increased year by year. Since yaks and some other livestock cannot walk into deep marsh, pastures are generally distributed in marshy meadow and meadow, and trenching activities for drainage have been rapidly supported by local herdsmen and have undergone vigorous development. A single ditch can cover some kilometers away. Changes in hydrologic conditions can directly affect the marsh wetland landscape. As marsh has degraded into marshy meadow, or even directly degraded into meadow, water conservation and ecological regulation capacity of the wetland has significantly declined which has led to further aggravation and irreversibility of the degradation process. Accordingly, between 1977 and 2016 a large area of marsh wetland has been drained on the Zoigê Plateau and grazing activities now encroach on the marsh core area. In many areas on the Zoigê Plateau, the dominant landscape has changed from an aquatic to a terrestrial landscape.

A county-level natural reserve was established in Zoigê County in 1994. In 1998, a national natural reserve (Zoigê Wetland National Natural Reserve) was established after approval by the State Council, this being recognized by the Ramsar Convention on Wetlands in the fourth batch of *Important List of National Wetland* in 2008. As noted, this natural reserve mainly aims to protect rare wildlife animals, such as the black-necked crane and the white stork, as well as the plateau marsh wetland ecological system. At the same time, the importance of the plateau wetland has been recognized and the degradation of the plateau marsh wetland has been identified as a significant issue; protection measures for alleviating degradation and the reduction of the marsh wetland have been implemented. Preliminary results (August 2016) for the protection of the Zoigê marsh wetland and the effect of remediation measures on the natural reserve indicate that the loss of marsh wetland reduced slowly. For the two types of marsh wetland landscape, the marsh landscape area began to increase and the fragmentation tendency was also effectively controlled.

## DISCUSSIONS

There are many studies on the evolution of wetland landscape patterns (*Cong et al., 2019*), most of which focused on the landscape pattern index such as TA, MPS, PLANT, LPI (*Ke et al., 2011*), or focused on the landscape fragmentation, landscape diversity, landscape uniformity and landscape dimension (*Tomaselli, Tenerelli & Sciandrello, 2012*). However, on the basis of the above landscape pattern analysis, this paper adds trend of centroid migration and slope with time. In addition, the related driving factors in marsh and marshy meadow were analyzed in this study. Previous studies have shown that the wetlands' degradation was closely correlated to the rise in air temperature, evaporation (*Bai, Lu & Wang, 2013*; *Bai et al., 2013*). As for the impact of human activities, the main concern is the impact of human activities on runoff, biogeochemical cycles (*Li et al., 2014*; *Chen et al., 2013*).

According to the conclusions we have obtained, the increase of annual average temperature was the natural factor while trenching and overgrazing were the main human factors resulting in wetland degradation. Therefore, if we want to protect the wetlands of the Zoigê Plateau, we must prohibit trenching, graze moderately, protect the environment, and slow down the warming of the climate.

However, this study has certain limitations in its research method and data selection. For example, the selected remote-sensing images in four different periods were not in different months, and the included cloud amount and shadow also imposed certain effects on the interpretation of results. All of these adverse factors can affect the accuracy in the calculation of marsh wetland landscape and landscape pattern indices. In the future, the relevant issues can be investigated in depth from the following aspects. The landscape pattern differentiation on the Zoigê Plateau can be examined on a smaller scale, and the variation tendencies of the marsh wetland landscape pattern on multiple scales can be compared. This will enable reasons for different variation tendencies of the landscape pattern indices to be provided and a full explanation for the ecological significance. It is advised that future studies begin from the micro-scale and analyze the coupling

relationship between ecological hydrological driving mechanisms of the marsh wetland and the macro-marsh landscape pattern.

## CONCLUSIONS

This study focused on landscape pattern change of marsh wetland in the Zoigê Plateau and analyzed the driving factors. The main conclusions from this study are:

(1) The marsh wetland area on the Zoigê Plateau has been reduced by approximately 66700 $hm^2$ since 1977, with a ratio of decline of 56.54%.

(2) The centroid of the marsh and marshy meadow landscape have the opposite trends, and the degradation of marsh wetland was more significant in the region where the angle of slope was greater.

(3) Currently, overall landscape pattern change of the marsh wetland on the Zoigê Plateau is characterized by a decrease in the degree of landscape fragmentation, diversity and fractal dimension.

(4) According to the analysis results of the driving forces, an increase in annual average temperature is the natural factor affecting wetland degradation and the excavation of trenches for drainage and overgrazing is the main human factor.

## ACKNOWLEDGEMENTS

We are very grateful to the editors and anonymous reviewers for their valuable comments, which greatly improved the quality of the paper.

### Funding

This work was funded by the Yunnan Provincial Department of Education Project (Grant 2019J0184, 2018JS346), the Yunnan Academician Workstation Project (Grant 2019IC012), the National Natural Science Foundation of China (Grant 41461022) and the Southwest Forestry University Research Startup Fund Project (Grant 111425). The funders had no role in study design, data collection and analysis, decision to publish, or preparation of the manuscript.

### Grant Disclosures

The following grant information was disclosed by the authors:
Yunnan Provincial Department of Education Project: 2019J0184, 2018JS346.
Yunnan Academician Workstation Project: 2019IC012.
National Natural Science Foundation of China: 41461022.
Southwest Forestry University Research Startup Fund Project: 111425.

### Competing Interests

The authors declare there are no competing interests.

## Author Contributions

- Liqin Dong conceived and designed the experiments, prepared figures and/or tables, authored or reviewed drafts of the paper, and approved the final draft.
- Wen Yang conceived and designed the experiments, authored or reviewed drafts of the paper, and approved the final draft.
- Kun Zhang conceived and designed the experiments, performed the experiments, authored or reviewed drafts of the paper, and approved the final draft.
- Shuo Zhen performed the experiments, prepared figures and/or tables, and approved the final draft.
- Xiping Cheng and Lihua Wu analyzed the data, prepared figures and/or tables, authored or reviewed drafts of the paper, and approved the final draft.

## Data Availability

The data are available as Supplementary Files.

## Supplemental Information

Supplemental information for this article can be found online at http://dx.doi.org/10.7717/peerj.9904#supplemental-information.

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
