# Peer review of "Study of marsh wetland landscape pattern evolution on the Zoigê Plateau due to natural/human dual-effects"

_PeerJ, doi:10.7717/peerj.9904_

## Round 0.1 · original submission · Major Revisions

I have received review comments from three experts in this field: one minor revision, one major revision, and one rejection. I have read the manuscript by myself and unfortunately, I agree with the comments of the reviewer who recommends rejection. However, I am happy to give your chance to improve the quality of the manuscript. If you can address all comments, especially the critical one, I will certainly reconsider my decision.

Reviewer 1 ·

Basic reporting

The authors analyzed marsh wetland landscape pattern evolution on the Zoigê Plateau from 1977-2016 by remote sensing and geographic information system technology. Generally the manuscript is well prepared. The authors provided relativelt sufficient field background. The conclusion can be supported by their results. And the the findings are important for wetland conservation and management.

Experimental design

This is an original work and the research question is well defined. The methodlogy in this stuy is reasonable and described in detail. .

Validity of the findings

The conclusions are well stated and can be supported by thier results.

You can add the alpine wetland protection policy based on your results to improve your article.

Additional comments

- Try not to use commas in the middle of the title.

- The article structure is suggested to be adjusted. Many data processes in the results parts, which should be moved to the method part. It’s also suggested that discussion and conclusion part should be separated, and it’s better to discuss first and then come to a conclusion.

- It is suggested to compare the results of this study with some similar studies which is done before (e.g., Effects of alpine wetland landscapes on regional climate,Advances in Meterology 2013), which can contribute to analyze the driving fators.

- You can add the alpine wetland protection policy based on your results to improve your article.

- There are some editting errors needed to be corrected before resubmission . For instance,

1) Lines 84-85: Rephrase, suggestion: “…, and a digital topographic map of the research area (with a resolution of 90 m) were used

2) Lines 221-223: Rephrase the sentence, suggestion: “…, which can also be confirmed by the above patch scale results. The correlations between…”

3) Lines 263-265: integrate the sentence, like this “…TA and PLAND declined gradually…”

Reviewer 2 ·

Basic reporting

This manuscript reports a landscape change during the passing decades for the Zoige Plateau. The authors present a clear background why the study is meaningful, and I agree with them. The English using is fine with grammar and spelling. The organization is also fine. However, the writing style is like a project report but not a scientific paper. For example, the result section is very redundant and can be reduced by 1/2. The discussion is fully of speculations, mostly without supporting evidence.

Experimental design

I have two major concerns:
1) There is no detail about how to distinguish the so-called swamp meadow and "mesophyte" meadow? --Moreover, I like using marshy meadow rather than swamp meadow because generally there is no trees or shrubs in the extensive meadow.

2) The authors attribute climate change and human disturbance to the change in landscape. This is perhaps a fact. However, they don't show evidence and they did not quantitatively determine to what extent these two factors influenced the landscape change.

Validity of the findings

Considering the vague description of the method section, I am not sure about the validity of the conclusion.

Additional comments

As noted above, this manuscript seems much immature to be a publication.

Reviewer 3 ·

Basic reporting

no comment

Experimental design

no comment

Validity of the findings

no comment

Additional comments

The thesis takes marsh wetland of Zoige Plateau as the research object, and analyzes the changes in the area and center of mass of marsh and swamp meadow in the past 40 years, as well as the evolution of landscape patterns, which are of great significance for wetland protection.
But the article is still inadequate, and it needs to be revised after being published.
(1) The discussion part needs to be strengthened and the latest research results need to be added;
(2) The legend in Figure 3 is marsh and marsh meadow. According to the description in the article, it should be marsh and swamp meadow, please check and correct it;
(3) The significance of studying the evolution of landscape pattern should be discussed in the introduction, please add it in the subsequent modification;
(4) In addition, there are some minor errors and irregularities in the article that need to be carefully checked, especially the usage of comma and definite article.
(5) I found some minor mistakes. Please check the manuscript carefully.
Line 199, tomographic should be topographic.
Line 227, variation should be various.
Line 347, of the local should be in the local.
Line 418, significances should be significance.

---

## Round 0.2 · Minor Revisions

Thanks you for your revision. As you can see, Reviewer 3 has two minor comments. Can you fix them before acceptance? Thanks.

Reviewer 1 ·

Basic reporting

The authors analyzed marsh wetland landscape pattern evolution on the Zoigê Plateau from 1977-2016 by remote sensing and geographic information system technology. The language of the revised manuscript is better than previous version. The authors provided a sufficient field background. The conclusion can be supported by their results. And the the findings are important for wetland conservation and management.

Experimental design

The experimental design is reasonable and the methodology was provided in details.

Validity of the findings

The conclusions are well stated and can be supported by their results.

Additional comments

The authors have solved all my concerns. I have no furhter comments on it. I recommend its acceptance for publication in the journal.

Reviewer 3 ·

Basic reporting

no comment

Experimental design

no comment

Validity of the findings

no comment

Additional comments

The authors have addressed most of my concerns. However, there still remain some issues that need to be resolved which are listed below.

1. Figure 1: Please improve the quality of the graph by adding DEM data as background. Please adding geographic grid and axes labels of longitude and latitude. I suggest adding a locator submap in Figure 1 to show location of Zoigê Plateau in China and Sichuan province.

2. Figure 3-9: Please save those figures in high resolution as it is pixelated. I suggest 800 dpi or greater.

---

## Round 0.3 · accepted · Accept

Thank you for submitting your manuscript to PeerJ. I think you have responded well to reviewers' comments, which was a minor revision after two rounds of reviews. Congratulations!